# Biosignal-Based Digital Biomarkers for Prediction of Ventilator Weaning Success

**DOI:** 10.3390/ijerph18179229

**Published:** 2021-09-01

**Authors:** Ji Eun Park, Tae Young Kim, Yun Jung Jung, Changho Han, Chan Min Park, Joo Hun Park, Kwang Joo Park, Dukyong Yoon, Wou Young Chung

**Affiliations:** 1Department of Pulmonology and Critical Care Medicine, Ajou University School of Medicine, Suwon 16499, Korea; petitprince012@ajou.ac.kr (J.E.P.); tomato81@aumc.ac.kr (Y.J.J.); lungmd@aumc.ac.kr (J.H.P.); parkkj@aumc.ac.kr (K.J.P.); 2BUD.on Inc., Jeonju 54871, Korea; tykim@bud-on.com; 3Department of Biomedical Systems Informatics, Yonsei University College of Medicine, Yongin 16995, Korea; changhohan8142@gmail.com (C.H.); jeff4273@yonsei.ac.kr (C.M.P.); 4Center for Digital Health, Yongin Severance Hospital, Yonsei University Health System, Yongin 16995, Korea

**Keywords:** weaning, prediction, mechanical ventilator, biosignal, machine learning, digital biomarker

## Abstract

We evaluated new features from biosignals comprising diverse physiological response information to predict the outcome of weaning from mechanical ventilation (MV). We enrolled 89 patients who were candidates for weaning from MV in the intensive care unit and collected continuous biosignal data: electrocardiogram (ECG), respiratory impedance, photoplethysmogram (PPG), arterial blood pressure, and ventilator parameters during a spontaneous breathing trial (SBT). We compared the collected biosignal data’s variability between patients who successfully discontinued MV (*n* = 67) and patients who did not (*n* = 22). To evaluate the usefulness of the identified factors for predicting weaning success, we developed a machine learning model and evaluated its performance by bootstrapping. The following markers were different between the weaning success and failure groups: the ratio of standard deviations between the short-term and long-term heart rate variability in a Poincaré plot, sample entropy of ECG and PPG, α values of ECG, and respiratory impedance in the detrended fluctuation analysis. The area under the receiver operating characteristic curve of the model was 0.81 (95% confidence interval: 0.70–0.92). This combination of the biosignal data-based markers obtained during SBTs provides a promising tool to assist clinicians in determining the optimal extubation time.

## 1. Introduction

Attempting to wean critically ill patients from mechanical ventilation (MV) is crucial. Reducing the duration of MV decreases ventilator-related pneumonia, muscle weakness, length of stay in the intensive care unit (ICU), and health care costs [1,2,3,4,5]. However, premature weaning may result in harmful outcomes such as complications during reintubation, deconditioning of the patient, an increased need for tracheostomy, and a potential increase in mortality [6,7,8,9,10,11,12]. Therefore, identifying the precise time for weaning from MV is a critical decision.

Although many predictive indices and clinical tools are already in use [1,13,14,15,16,17,18,19,20,21,22], >20% of the patients who have fulfilled the classic weaning criteria require reintubation [14,23,24,25]. The predictive performance decreases in patients with multi-organ dysfunction, older age, prolonged MV, and severe illness [26,27,28,29,30,31]. The lack of reliable weaning parameters is related to the heterogeneity of critically ill patients and their ever-changing clinical courses [32,33,34,35]. The causes of weaning failure are not exclusively attributable to oxygenation or ventilation insufficiency; cardiac function, volume status, muscle deconditioning, and the presence of delirium also affect weaning outcomes [9,23,36,37,38,39,40]. Most of the indices are based on the clinical situation recorded at a single time point, although each patient’s oxygenation, ventilation, hemodynamic, musculoskeletal, and mental statuses are often unstable and vary over time.

In this study, we hypothesized that features extracted from biosignals collected during the weaning process would provide better predictive information than the commonly used rapid shallow breathing index (RSBI); specifically, biosignal-based features would include more diverse physiological information regarding the patient’s status (i.e., information not limited to the pulmonary system). This hypothesis has been supported by recent studies, in which useful digital biomarkers were present in biosignal data [26,41,42,43,44]. These markers may predict or detect cardiovascular and other clinical events [45,46,47]. Electrocardiogram (ECG) data reflect heart status, and specific morphology of the arterial blood pressure waveform could reflect the status of the cardiovascular system [48,49,50,51,52]. Photoplethysmogram (PPG) data are used to measure oxygen saturation and provide information regarding oxygen transfer [53,54,55,56]. A healthy biosystem is characterized by complexity and variability, and alterations in variability and reduced complexity are related to pathological conditions [57,58,59,60,61,62]. For ventilator weaning, breathing pattern variability analysis has been performed for the estimation of weaning readiness in many studies; reductions in variability indices during a spontaneous breathing trial (SBT) were reportedly associated with extubation failure [63,64,65,66,67].

Here, we proposed a biosignal-based weaning prediction approach, which would continuously reflect the patient’s clinical and physiological progression over time. This study aimed to compare the distribution of values of biosignal data between the weaning success and failure groups during an SBT. It also aimed to evaluate the additional value of the biosignal data for the prediction of extubation outcomes, compared with the commonly used RSBI.

## 2. Materials and Methods

This retrospective study was conducted using anonymized data. The Institutional Review Board of Ajou University Hospital approved the study (IRB No. AJIRB-MED-MDB-20-090) and waived the requirement for informed consent.

### 2.1. Data Sources

We collected clinical and biosignal data from patients who were admitted to the ICU and underwent MV at Ajou University Hospital, a tertiary teaching hospital in South Korea, from January 2019 to November 2020. Clinical data obtained from electronic medical records and biosignal data were collected using our custom biosignal collecting platform, which we developed for research purposes [68]. We also collected ventilator parameters directly from the ventilators following every SBT to accurately identify the breathing patterns and their variability (Figure 1).

### 2.2. Study Population

Patients aged ≥ 18 years who had undergone MV for at least 24 h and who fulfilled the weaning criteria were included (Figure 2). Weaning criteria were applied according to our institution’s ventilator weaning protocol, which was based on the guidelines developed by the American College of Chest Physicians, the American Association for Respiratory Care, and the American College of Critical Care Medicine, with reference to additional research methods [1,10,69,70]. The weaning criteria were as follows: resolution or improvement of the condition leading to intubation; hemodynamic stability, which was defined as systolic blood pressure between 90 and 160 mmHg, and heart rate <140 beats/min with low or no doses of vasopressors; stable neurological status (no deterioration in Glasgow Coma Scale during the prior 24 h); respiratory stability (oxygen saturation >90% with fraction of inspired oxygen [FiO_2_] ≤ 0.4), respiratory rate < 35/min, spontaneous tidal volume >5 mL/kg; and intact cough and gag reflexes. Patients with a tracheostomy or a do-not-reintubate order were excluded.

All the patients underwent a 30-min SBT with ≤6 cm H_2_O pressure support ventilation and positive end-expiratory pressure; the FiO_2_ remained unchanged from the MV period prior to the SBT. When the patients successfully passed the 30-min SBT, they were extubated and provided with a high-flow nasal cannula or air entrainment mask for oxygen therapy. Patients who did not tolerate the SBT were reconnected to a ventilator. The criteria for failure to tolerate the SBT were agitation, anxiety, deterioration of consciousness, respiratory rate > 35/min and/or use of accessory muscles, oxygen saturation by pulse oximetry <90% with FiO_2_ > 0.5, heart rate > 140/min or >20% increase from baseline, systolic blood pressure <90 mmHg, or development of an arrhythmia.

### 2.3. Study Design

This study focused on the variability of the physiological responses to the following abrupt changes in the external environment: support with MV, reduced ventilator support, and increased respiratory demand because of the SBT. To compare the biosignal features between the weaning success and failure groups, we defined the two groups (as described below), and then calculated the biosignal features representing those variabilities (Figure 2).

We defined the weaning failure group as patients who failed to wean before extubation and patients who were reintubated within 48 h following extubation. Failed extubation was defined as reintubation within 48 h of extubation. Respiratory failure within 48 h of extubation was defined as the occurrence of at least one of the following: respiratory acidosis with pH <7.3 and partial pressure of carbon dioxide (PaCO_2_) >45 mmHg, oxygen saturation <90% with FiO_2_ >0.5, respiratory rate >35/min, deterioration of consciousness, severe agitation, or clinical signs of respiratory fatigue. We defined the extubation success group as patients free from MV for >48 h following extubation. Two pulmonologists (W.Y.C. and J.E.P.) reviewed the clinical data of all the enrolled patients and confirmed whether the patients were included in the case (success) or control (failure) groups.

### 2.4. Feature Extraction

To extract biosignal-based features, we used waveform data including ECG, PPG, respiratory impedance, and invasive arterial blood pressure measurements, as well as numerical measurements including heart rate, respiratory rate, and mean arterial pressure. All the waveform data were down-sampled as 62.5 Hz to ensure that they have the same data format and to reduce computational complexity in the analysis. We also collected and used ventilator parameters for every breath from mechanical ventilators including tidal volume, inspiration time, and the ratio of inspiratory and expiratory time during the 30-min SBT.

We calculated the features that represented the variability of time-series data using Poincaré plots, sample entropy (SampEn), and detrended fluctuation analysis in the middle 10 min of the SBT.

A Poincaré plot is a scatter plot of the current value (e.g., the R-R interval in an ECG) against the immediately preceding value (Figure 3A). Standard deviation 1 (SD1) in the plot is defined as the level of deviation against the line of identity (y = x). SD1 represents how consecutive values differ from previous values (short-term variability). SD2 is calculated as the level of deviation together with the line of identity (i.e., how all values are distributed; long-term variability). The ratio of SD1 and SD2 (SD1/SD2) represents the level of short-term variability, compared with long-term variability.

SampEn is an index that represents the level of complexity of a particular dataset (Figure 3B). It calculates the probability that the same findings are observed in different time windows; the calculated value is then used as input in a negative logarithm. A low SampEn value indicates a high level of regularity; a high SampEn value indicates an irregular state.

Detrended fluctuation analysis is used to quantify the level of fractal-like correlation of the time-series data. When the patterns observed in some time windows are also observed in the larger or smaller time windows, this is regarded as fractal-like correlation. An α = 0.5 indicates random data with no pattern, while α >0.5 indicates data with fractal correlation. Usually, two indicators are calculated, α1 and α2, indicating short- and long-term fractal-like correlation (fluctuation) (Figure 3C).

### 2.5. Statistical Analyses

Categorical variables are presented as numbers and percentages. Continuous variables are summarized as means and standard deviations. To compare categorical variables, the χ2 test or Fisher’s exact test was used. Mann–Whitney U test was used for continuous variables. In the comparison of the baseline characteristics between the success and failure group, *p* < 0.05 was considered significant. In the comparison of the biosignal-based features, *p* < 0.1 was used to include more diverse variables as input values for the machine learning model for weaning prediction. However, to control the false discovery rate owing to multiple comparisons, we used the Benjamini and Hochberg correction. Thresholds for statistical significance α_adj_ were adjusted as α**i*/*m*, where α = 0.1, *m* is the number of comparisons, and *i* is the position in an ordered *p*-value list from smallest to largest (1, …, *m*). Statistical significance was defined as *p* < α_adj_.

### 2.6. Development of the Machine Learning Model

To evaluate whether the composite of biosignal-based features is useful to predict the probability of weaning success, we developed a machine learning model using biosignal features. We included not only biosignal features that showed significant differences but also features that showed near significant differences between the case and control groups, to include all variables that could have additional information in weaning prediction. The RSBI value, which is currently used for weaning prediction, was also included in the input values.

A Random Forest classifier was used to predict the weaning failure. The Random Forest classifier is consisted of series of independent decision trees. Each tree has a hierarchical decision rules, and it separates input data into *N*(*t*) leaves ∈ [*N_t_*_,1_, …, *N_t,N_*_(*t*)_], where t ∈ (1, …, *T*) means each tree and *N_t,i_* contains a probability of weaning success π_t,i_ ∈ [0,1]. The Random Forest model collects all prediction from each of decision models, and it returns the majority of votes as final output. The character of the classifier can be determined by hyperparameters. In this study, the following hyperparameters were set: the function to measure the quality of a split was “Gini Impurity,” the number of estimators was 50, the minimum number of samples required to constitute a leaf node was one, and the minimum number of samples required to split an internal node was two. The Gini Impurity was calculated using the following formula:$$G=\overset{C}{\underset{i=1}{\sum }}p\left(i\right)\times \left(1-p\left(i\right)\right)$$
where *C* is the number of total classes and *p*(*i*) is the probability of selecting data with class *i* (weaning success or failure). Each internal node was trained to have best splits the space of training data to lead to the greatest reduction in Gini Impurity defined above.

We determined the relative importance of the features after training. The feature importance of a Random Forest indicates the degree to which the overall classification impurity is reduced if the feature is used in the model. To calculate importance of each feature, the importance of each node was calculated using the following formula:$$ni_{j}=w_{j}G_{j}-w_{left\left(j\right)}G_{left\left(j\right)}-w_{right\left(j\right)}G_{right\left(j\right)}$$
where *ni_j_*, *w_j_*, *G_j_*, *left*(*j*), *right*(*j*) means the importance of node *j*, weighted number of samples reaching node *j* (*N_j_*/*N*), the impurity value of node *j*, child node from left split on node *j*, child node from right split on node *j*, respectively. Further, the importance of each feature was then calculated using the following formula:$$fi_{i}=\frac{\sum_{j\in nodes\;split\;on\;feature\;i}ni_{j}}{\sum_{k\in all\;nodes}ni_{k}}$$
where *fi_i_* and *n_i_* means the importance of feature *i* and node *j*. Then *fi_i_* was normalized by dividing by the sum of all feature importance value to make all *fi_i_* values be ranged between 0 and 1. Finally, the averaged *fi_i_* over all trees was used to evaluate *fi_i_* at the random forest level.

Performance of the model was measured by sensitivity, specificity, positive predictive value, negative predictive value, F1 score, and area under the receiver operating characteristic (AUROC). To compare the usefulness of our model with the existing RSBI alone, the weaning prediction performance of the RSBI was evaluated using the same performance measures. The dataset was randomly separated into training and test datasets in the ratio of 7:3. The average performance and 95% confidence intervals of performance indices were calculated using bootstrap resampling procedures with 1000 iterations. For performance comparison, we also conducted multiple logistic regression with the same process.

### 2.7. Software Used in the Study

Acquisition software (Hamilton Medical Ventilator data logger version 5.0, Bonaduz, Switzerland) was used to obtain the numerical data, such as tidal volume, inspiratory time, and the ratio of inspiratory and expiratory time from the mechanical ventilators, during the trial. Microsoft SQL Server and Python were used for data management and statistical analyses.

## 3. Results

During the study period, 350 patients underwent MV in the ICU. Of these 350 patients, 106 fulfilled the inclusion criteria, 17 were excluded due to missing biosignal or ventilator data, and 89 were finally included in the study (Figure 2). Among the 89 included patients, 67 successfully discontinued MV and were able to breath by themselves without the aid of a ventilator for at least 48 h following extubation (case group); and 22 patients failed and resumed MV within 48 h (control group). The baseline characteristics of the two groups are provided in Table 1. They were similar in terms of age, sex, main cause of ICU admission, APACHE II score, and length of MV before the SBT. In particular, in the case of pneumonia, which accounts for a large proportion of the reasons for ICU admission, there was no difference between the two groups in the comparison according to the type of pneumonia and the causative pathogen.

Among the biosignal-based features, we could detect significant differences between the two groups in the following features (Table 2): SampEns in ECG and PPG. We also included the additional following variables that showed near significant level of difference for further analysis (machine learning model development): ratio of SD2 and SD1 in heart rate; ɑ1 values in ECG and respiratory impedance, ɑ2 values and ɑ1/ɑ2 in ECG.

When the RSBI value alone was used for weaning prediction, its AUROC was 0.58 (95% confidence interval: 0.44–0.71). When the biosignal-based features were combined with RSBI for weaning prediction, the AUROC value increased to 0.81 (95% confidence interval: 0.70–0.92). The performance comparison of the models (based on biosignals with the RSBI score and based on RSBI alone) is shown in Table 3. The combined model demonstrated improved specificity (+26%), accuracy (+4%), negative predictive value (+18%), and F1 score (+23%), with a similar level of sensitivity. The details of model performance are provided in Table 3 and Figure 4.

As shown in Figure 5, biosignal-based features selected in this study exhibited value similar to the existing RSBI for predicting weaning success in the Random Forest model. In particular, SD1/SD2 in heart rate, α1 in respiration impedance, and SampEn in PPG were more valuable than RSBI.

## 4. Discussion

Our study successfully incorporated novel biosignal-based features into classic weaning prediction tools to provide a more accurate marker for MV discontinuation. SD1/SD2 in heart rate, SampEn in ECG, α1 in respiratory impedance, and SampEn in PPG were significant discriminants of MV weaning success. The addition of this panel of new parameters to the RSBI yielded better predictive performance, compared with RSBI alone.

During the weaning process, we observed differences between the success and failure groups based on the biosignal features extracted from heart rate, ECG, respiratory impedance, and PPG. In the analysis of numerical data, heart rate distribution using the Poincaré plot method for patients with weaning failure showed reduced variability of measured parameters. A decrease in variability reportedly indicates reduced adaptive capacity in a stressful environment (e.g., reduced ventilator support) and has been described in several pathological conditions [71,72]. Heart rate variability (HRV) is the time interval between consecutive heartbeats, and is a commonly used variable for predicting weaning outcomes using biosignal data analysis. HRV is associated with the balance between parasympathetic and sympathetic regulation, thermoregulation, baroreflexes, and respiration; altered HRV is reportedly associated with failed weaning trials [46,73]. Huang et al. reported that when analyzing HRV in the pre-SBT, SBT, and post-extubation periods, decreased HRV was significantly associated with SBT failure; the inability to increase HRV following extubation was correlated with subsequent reintubation [74]. Seely et al. also reported that alterations in the HRV during SBTs significantly correlated with weaning failure [64]. The weaning process is associated with increased breathing effort; Seely et al. suggested that the inability to tolerate the increased breathing effort in patients who were not ready for extubation could be used to improve the prediction of failed extubation. In our study, we used numerical values of heart rate per minute measured every second, which would be difficult to compare directly using HRV. However, the increased variability of the cardiovascular system to adapt to environmental changes is presumably through a similar mechanism.

In analysis of the waveform data, patients who were successfully extubated had lower complexities in the ECG, respiratory impedance, and PPG during the SBTs. We presume that patients with weaning success exhibited better preservation of regularity and reproducibility in biosignal features, compared with patients with weaning failure; patients with weaning success presented low complexity and predictable features. In contrast, the biological rhythms became more irregular and unpredictable in patients who did not tolerate the weaning process in our study. Engoren et al. reported that the weaning failure group showed increased irregularity in biosignal analysis of approximate entropy of tidal volume, which reflects enhanced external inputs to the respiratory control center; increased regularity in the weaning success group indicated a better adaptive mechanism of an autonomous system [75]. El Khatib et al. reported that Kolmogorov entropy and dimensions of the spontaneous breathing pattern were increased in patients who failed weaning trials; they also suggested that complexity during the SBTs was enhanced in patients with weaning failure [41]. In another study, Papaioannou et al. assessed the respiratory pattern complexity in critically ill surgical patients during weaning trials; they reported that patients with weaning failure exhibited significantly decreased respiratory pattern complexity, reduced SampEn, and increased detrended fluctuation analysis exponents, compared with patients with weaning success [76]. Discrepancies in the results, compared with previous studies, are presumably associated with different protocols for weaning and different patient characteristics. Papaioannou et al. compared the before and after SBTs in both successful and unsuccessful groups. Our study directly compared the biosignal features of the weaning success and failure groups. We suspect that the success group showed predictable variability owing to control by the internal regulatory system, while the failure group exhibited a more chaotic behavior since these patients were unable to tolerate environmental changes.

In this study, we also analyzed the variability and complexity of respiratory rate, tidal volume, arterial blood pressure, and inspiratory to expiratory time ratio (I:E ratio) over time using the machine learning model; however, they failed to show any significant results in our study.

Respiratory rate and tidal volume are the main variables of RSBI, which is the most coveted weaning predictor in the ICU so far, and their stability were expected to demonstrate a significant role in the prediction of weaning success. Their limited role in our study could be because once they are maintained under a certain value, the change over time may have little significance. A ratio of respiratory rate over tidal volume (i.e., RSBI) under 105 is considered adequate for considering extubation in many critical care guidelines, and if the value is maintained under 105, the stability should not be a major factor in weaning success [77,78]. Moreover, since the values are included in the RSBI itself, the comparison of these biosignals with RSBI would not incur any difference. Furthermore, in order to collect the ventilator associated biosignals, we provided minimal ventilatory support until extubation, instead of disconnecting MV during SBTs. This should have maintained stable tidal volume to maintain the volume over the minimally required value to reduce the predictive performance of the breathing pattern variability. According to Otaguro et al. respiratory rate and tidal volume were the least important weaning predictors in their study comparing different machine learning models for successful extubation prediction [79].

Similar to heart rate, blood pressure reflects the patient’s cardiovascular reserve. Its instability over time is more a matter of poor cardiovascular function or inadequate volume status than the patient’s adaptation mechanism to decreasing ventilatory support. Before selecting weaning candidates, we carefully achieved optimal volume control and checked their sufficient cardiovascular function in order to protect patients from negative cardiovascular events during the weaning procedure. Inspiratory time to expiratory time ratio in self-breathing patients is a function of respiratory system resistance [80]. Therefore, before undergoing the weaning procedure, it is fundamental to control pathologic conditions, which can increase the bronchial resistance, such as bronchial spasm or exacerbation of airway disease. This conservative approach is endowed with a low discriminating value to inspiratory time complexity as well as I:E ratio variability.

The prediction of weaning outcome improved with the combination of biosignal markers and RSBI, compared with RSBI alone. RSBI is derived from the respiratory frequency divided by tidal volume, and thus directly represents breathing characteristics. Furthermore, RSBI is the most used index for the estimation of weaning readiness during an SBT [77,78]. If the patient has adequate tidal volume with deep and regular breathing, the RSBI will be low, which suggests weaning success. However, the purposes of breathing are to successfully inhale air into the lungs and transfer oxygen to each tissue in the body. Breathing quality may be affected by many diverse variables, including the cardiovascular system, autonomic nervous system, and musculoskeletal capacity [81,82]. In this study, the prediction model that integrated RSBI and biosignal data demonstrated higher values of performance indices, compared with RSBI alone.

Previous studies have suggested that clinically important information remained undiscovered in the biosignal data [83,84]. In the ECG waveform data, atrial fibrillation could be detected regardless of whether the ECG waveforms maintained normal sinus rhythm [48]. Normal sinus rhythm (presumed to indicate completely normal status) is expected to include information for mortality prediction [49]. Morphological changes of P waves suggest an increased risk of hemorrhage in patients with ischemic stroke [50]. ECG waveform information is also useful when screening for cardiac contractile dysfunction [51,85]. Our results in this study suggest that biosignal data (e.g., ECG, PPG, and arterial blood pressure) provide useful information, specifically regarding whether patients attempting an SBT have sufficient ability to breath without MV assistance.

Incorporation of biosignal-based features when monitoring patients during the weaning process enables physicians to make rapid clinical decisions based on real-time, continuous medical information [86,87,88]. Continuous monitoring of biosignal information, such as ECG, respiratory rate, PPG, and arterial blood pressure is routine practice in ICU care. Thus, biosignal-based measurements are easily accessible and would be helpful when assessing patients attempting SBTs. Our model did not use any clinical information other than the biosignal data obtained from the patient monitoring devices. Therefore, our model can easily be used in the ICU setting.

A limitation of this study was its single-center design (i.e., single ICU at a single institute). We did not include patients admitted for surgical or trauma-related ICU care, owing to their ventilator use characteristics. Post-surgical use of MV in the ICU rarely causes weaning failure associated with a primary problem in the pulmonary system. We included only patients who received respiratory support with MV because of respiratory problems, while excluding other ICU patients. The medical records of all enrolled patients were reviewed by two board-certified pulmonologists. Owing to this process, we found no significant differences in the baseline characteristics between the weaning success and failure groups.

The small number of patients was another limitation in this study, although we identified meaningful biosignal-based digital biomarkers. Previous studies discovered novel biosignal-based digital biomarkers using deep learning [41,42,64,75]. Current deep learning techniques can identify hidden patterns with large amounts of data; however, we could not collect sufficient data to support a deep learning model. We presume that further valuable information for determining the possibility of weaning success can be discovered when more data are obtained, supporting the development of a deep learning model.

## 5. Conclusions

MV weaning failure is usually multifactorial; thus, the weaning parameters that assess a single physiological function may have limited predictive accuracy. In the weaning process, changes in biosignal markers can serve as predictive indicators of each patient’s extubation outcome. These changes offer a noninvasive and valuable tool to characterize cardiorespiratory function and autonomic system interactions. We identified a new biosignal-based combination of markers to determine the possibility of weaning success. By using these digital biomarkers, clinicians can select the appropriate earliest weaning time, which could decrease the risks of both unnecessarily prolonged ventilator support and premature weaning. Therefore, new marker-based biosignal data obtained during SBTs provide a promising tool to assist clinicians in determining the optimal extubation time for the treatment of critically ill patients undergoing ventilator care. However, confirmation of the model generalizability warrants additional studies in other institutions to obtain larger numbers of patients.

## Figures and Tables

**Figure 1 ijerph-18-09229-f001:**
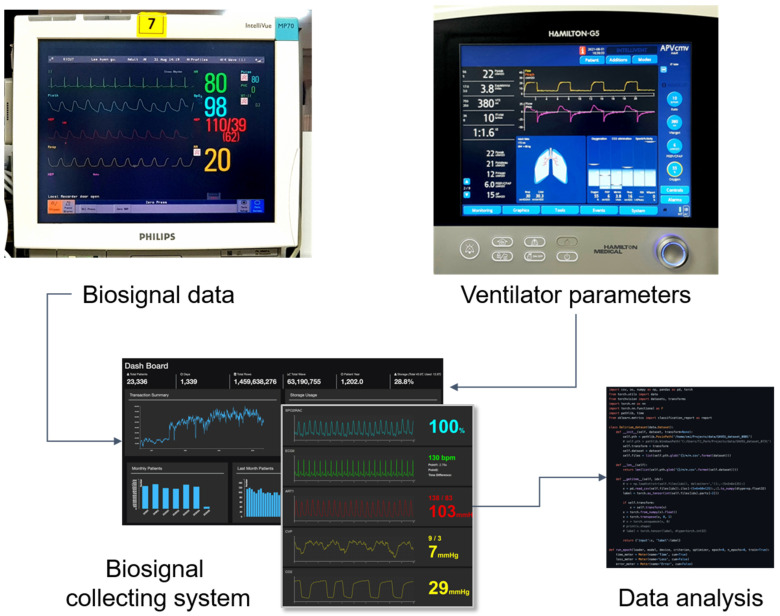
Biosignal data collecting and data analysis process. Biosignal data from patient monitor devices and parameters from mechanical ventilators were collected via our biosignal collecting system. Collected data were analyzed retrospectively to find out features for predicting weaning success.

**Figure 2 ijerph-18-09229-f002:**
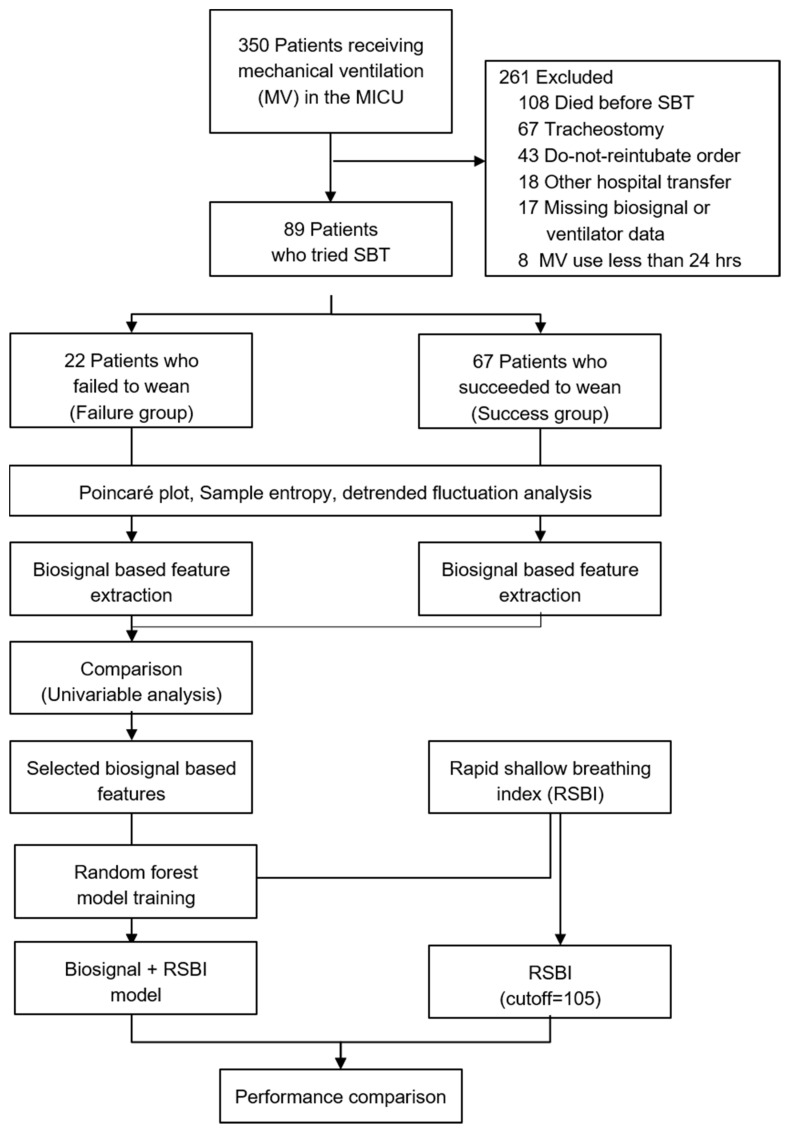
Study overview. We selected biosignal-based features that showed differences between the weaning failure and success groups. Their usefulness was evaluated by applying these features to predict weaning success, followed by comparison with the pre-existing RSBI. ICU, intensive care unit; RSBI, rapid shallow breading index; SBT, spontaneous breathing trial.

**Figure 3 ijerph-18-09229-f003:**
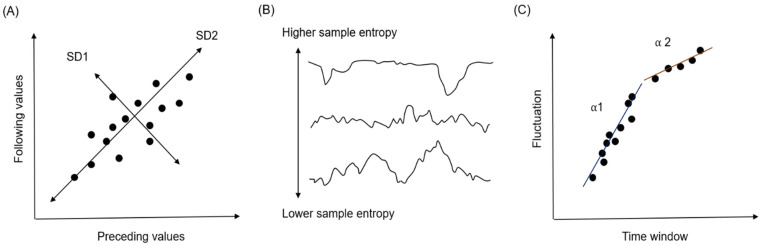
Methods used for the feature extraction process. (**A**) Poincaré plot, (**B**) Sample entropy, and (**C**) Detrended fluctuation analysis. All three approaches evaluate the level of variability in time-series data. For numerical values, the Poincaré plot method was used; for waveform data, sample entropy and detrended fluctuation analysis methods were used.

**Figure 4 ijerph-18-09229-f004:**
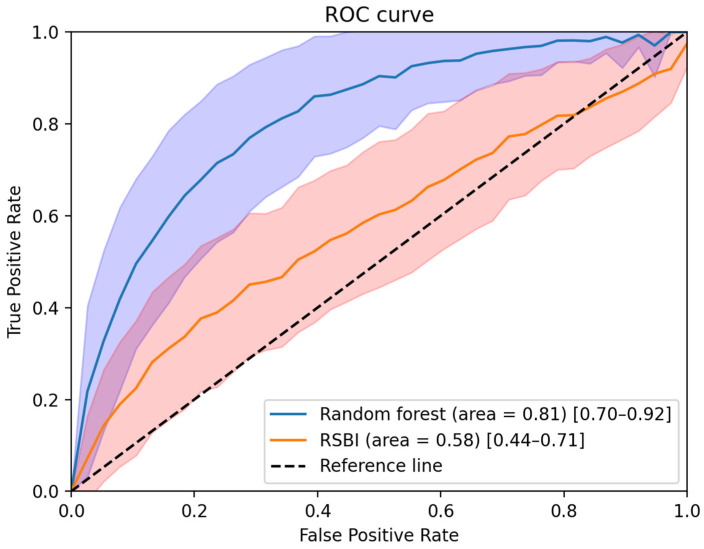
Performance comparison between the model using RSBI alone and the model using RSBI and biosignal-based features. After 1000 iterations of bootstrapping, the mean AUROC and 1 standard deviation of each group are shown as a solid line and shaded area.

**Figure 5 ijerph-18-09229-f005:**
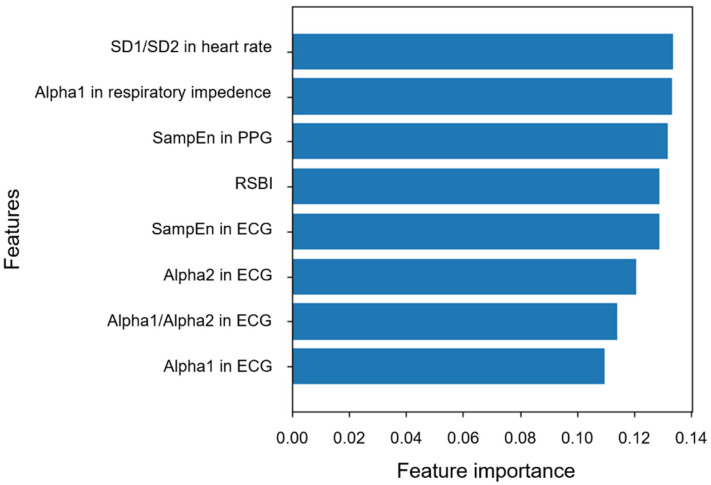
Feature importance in the Random Forest model to predict weaning success. All biosignal features were demonstrated to have value similar to RSBI.

**Table 1 ijerph-18-09229-t001:** Baseline characteristics of the study population according to the outcome of weaning.

Characteristics	Total (*N* = 89)	Success Group (*N* = 67)	Failure Group (*N* = 22)	*p* Value
Age, mean ± SD, year	69.3 ± 14.3	69.8 ± 13.5	67.59 ± 16.5	0.533
Sex (males/females), *n*	54/35	40/27	14/8	0.743
Body weight, mean ± SD, kg	59.2 ± 11.7	59.6 ± 12.2	57.9 ± 10.3	0.568
Height, mean ± SD, cm	164.5 ± 9.6	163.8 ± 10.0	166.8 ± 8.0	0.193
BMI, mean ± SD, kg/m^2^	21.9 ± 4.2	22.3 ± 4.3	20.8 ± 3.7	0.152
Main cause of ICU admission, *n*(%)				0.897
Pneumonia	59 (66.3)	45 (67.2)	14 (63.6)	
COPD/Asthma AE	8 (9.0)	6 (9.0)	2 (9.1)	
Pulmonary hemorrhage	4 (4.5)	3 (4.5)	1 (4.5)	
Sepsis	3 (3.4)	3 (4.5)	0 (0)	
Gastrointestinal bleeding	1 (1.1)	1 (1.5)	0 (0)	
Neurologic disease	2 (2.2)	1 (1.5)	1 (4.5)	
Pulmonary edema	7 (7.9)	5 (7.5)	2 (9.1)	
Others	5 (5.6)	3 (4.5)	2 (9.1)	
Comorbidity, *n*(%)				
Cardiovascular disease	52 (58.4)	40 (59.7)	12 (54.5)	0.670
Diabetes mellitus	25 (28.1)	20 (29.9)	5 (22.7)	0.519
Chronic obstructive pulmonary disease	16 (18.0)	11 (16.4)	5 (22.7)	0.530
Neurological disease	24 (27.0)	20 (29.9)	4 (18.2)	0.285
Malignancy	18 (20.2)	14 (20.9)	4 (18.2)	>0.99
Renal disease	10 (11.2)	9 (13.4)	1 (4.5)	0.440
Liver disease	4 (4.5)	4 (6.0)	0 (0)	0.568
APACHE II score, mean ± SD	21.8 ± 8.1	22.3 ± 8.3	20.2 ± 7.3	0.288
Length of mechanical ventilation before SBT, mean ± SD, d	7.3 ± 5.3	7.0 ± 5.5	8.1 ± 4.7	0.393
Duration of MV ≥ 72 h, *n*(%)	68 (76.4)	50 (74.6)	18 (81.8)	0.491
Use of neuromuscular blocker, *n*(%)	18 (20.2)	13 (19.4)	5 (22.7)	0.764
Excess secretion, *n*(%)	9 (10.1)	6 (9.0)	3 (13.6)	0.684
Arterial blood gas ananlysis, mean ± SD				
PaO_2_, mmHg	107.8 ± 34.8	108.0 ± 31.4	106.9 ± 44.4	0.891
PaCO_2_, mmHg	38.7 ± 11.0	37.6 ± 10.4	41.9 ± 12.4	0.116
PaO_2_/FiO_2_ ratio	317.3 ± 102.3	320.5 ± 91.4	307.3 ± 132.1	0.666
Upper airway disorder after extubation, *n*(%)	2 (2.2)	2 (3)	0 (0)	>0.99
Prior failed weaning attempt, *n*(%)	15 (16.9)	9 (13.4)	6 (27.3)	0.187

Data are presented as mean ± standard deviation or number (%). BMI, body mass index; ICU, intensive care unit; COPD, chronic obstructive pulmonary disease; AE, acute exacerbation; APACHE, acute physiology and chronic health evaluation; SBT, spontaneous breathing trial; MV, mechanical ventilation; PaO_2_, partial pressure of oxygen in arterial blood; PaCO_2_, partial pressure of carbon dioxide; FiO_2_ ratio, fraction of inspired oxygen.

**Table 2 ijerph-18-09229-t002:** Univariate analysis results of biosignal features between the weaning success and failure groups.

Items	Variability Index	Success Group	Failure Group	*p* Value	α_adj_ ^†^
Heart rate	SD1 (mean ± SD)	2.52 (1.45)	2.17 (1.15)	0.316	0.035
	SD2 (mean ± SD)	6.74 (4.56)	8.63 (6.98)	0.741	0.076
	SD1/SD2 (mean ± SD)	0.43 (0.18)	0.32 (0.13)	0.015 *	0.009
Respiratory rate	SD1 (mean ± SD)	2.76 (1.22)	1.77 (0.49)	0.617	0.068
	SD2 (mean ± SD)	2.94 (1.25)	3.08 (1.25)	0.561	0.059
	SD1/SD2 (mean ± SD)	0.62 (0.16)	0.65 (0.24)	0.592	0.065
Tidal volume	SD1 (mean ± SD)	52.58 (32.96)	27.64 (10.9)	0.237	0.029
	SD2 (mean ± SD)	72.90 (40.18)	45.40 (9.86)	0.747	0.079
	SD1/SD2 (mean ± SD)	0.72 (0.2)	0.62 (0.19)	0.496	0.053
IE ratio	SD1 (mean ± SD)	61.23 (47.58)	164.57 (297.75)	0.882	0.094
	SD2 (mean ± SD)	61.23 (47.58)	189.32 (268.39)	0.408	0.044
	SD1/SD2 (mean ± SD)	0.63 (0.22)	0.55 (0.23)	0.318	0.038
Inspiratory time	SD1 (mean ± SD)	96.20 (68.04)	79.98 (53.70)	0.750	0.082
	SD2 (mean ± SD)	146.79 (79.75)	156.47 (117.67)	0.567	0.062
	SD1/SD2 (mean ± SD)	0.66 (0.24)	0.58 (0.28)	0.511	0.056
Mean ABP	SD1 (mean ± SD)	5.37 (4.58)	5.99 (9.42)	0.340	0.041
	SD2 (mean ± SD)	10.80 (6.09)	15.98 (23.12)	0.832	0.088
	SD1/SD2 (mean ± SD)	0.5 (0.21)	0.4 (0.17)	0.087	0.026
ECG	SampEn (mean ± SD)	2.04 (0.61)	2.50 (0.46)	0.005 **	0.006
	ɑ1 (mean ± SD)	1.29 (0.15)	1.22 (0.08)	0.033 *	0.021
	ɑ2 (mean ± SD)	0.57 (0.23)	0.43 (0.20)	0.016 *	0.012
	ɑ1/ɑ2 (mean ± SD)	2.83 (2.46)	3.47 (1.57)	0.026 *	0.018
Respiratory impedance	SampEn (mean ± SD)	0.21 (0.05)	0.22 (0.05)	0.413	0.047
	ɑ1 (mean ± SD)	2.03 (0.04)	2.01 (0.03)	0.018 *	0.015
	ɑ2 (mean ± SD)	1.11 (0.27)	1.07 (0.31)	0.719	0.074
	ɑ1/ɑ2 (mean ± SD)	2.04 (1.11)	2.70 (3.64)	0.973	0.1
PPG	SampEn (mean ± SD)	0.14 (0.05)	0.18 (0.11)	0.002 **	0.003
	ɑ1 (mean ± SD)	1.96 (0.12)	1.91 (0.19)	0.062	0.024
	ɑ2 (mean ± SD)	1.96 (0.12)	0.78 (0.49)	0.429	0.05
	ɑ1/ɑ2 (mean ± SD)	5.61 (10.72)	−2.15 (29.37)	0.947	0.097
ABP	SampEn (mean ± SD)	0.36 (0.39)	0.43 (0.46)	0.247	0.032
	ɑ1 (mean ± SD)	2.09 (0.01)	2.08 (0.02)	0.627	0.071
	ɑ2 (mean ± SD)	1.83 (0.10)	1.82 (0.12)	0.775	0.085
	ɑ1/ɑ2 (mean ± SD)	1.14 (0.07)	1.15 (0.08)	0.853	0.091

Data are presented as mean ± standard deviation. IE ratio, inspiratory to expiratory time ratio; ABP, arterial blood pressure; ECG, electrocardiogram; PPG, photoplethysmogram; SD1, standard deviations between short-term heart rate variability; SD2, standard deviations between long-term heart rate variability; SampEn, sample entropy. ^†^ α_adj_ is the adjusted statistically significant threshold α = 0.1 by multiple testing by Benjamini and Hochberg (BH) correction. ** Statistically significant variables after BH correction. * Statistically near significant variables after BH correction.

**Table 3 ijerph-18-09229-t003:** Performance indices (accuracy, sensitivity, specificity, positive predictive value, negative predictive value, and F-1 score).

	Sensitivity	Specificity	Accuracy	PPV	NPV	F-1 Score
RSBI (≥105)	0.91 (0.87–0.96)	0.26 (0.13–0.38)	0.80 (0.75–0.84)	0.85 (0.83–0.88)	0.40 (0.20–0.61)	0.30 (0.17–0.44)
RSBI + biosignal (Random Forest)	0.91 (0.85–0.97)	0.52 (0.36–0.69)	0.84 (0.79–0.89)	0.90 (0.87–0.93)	0.58 (0.40–0.76)	0.53 (0.40–0.66)
RSBI + biosignal (Multiple regression)	0.91 (0.86–0.97)	0.41 (0.25–0.57)	0.82 (0.78–0.87)	0.88 (0.85–0.91)	0.53 (0.33–0.73)	0.44 (0.30–0.58)

Mean values of areas under the curve with 95% confidence intervals. PPV, positive predictive value; NPV, negative predictive value; RSBI, rapid shallow breathing index.

## Data Availability

The data presented in this study are available on request from the corresponding author.

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
