# Peer review of "Biosignal-Based Digital Biomarkers for Prediction of Ventilator Weaning Success"

_ijerph, 2021, doi:10.3390/ijerph18179229_

Round 1

Reviewer 1 Report

Dear Authors, 

Thank you for sending this excellent manuscript for review.
I have a few comments on the manuscript.

The abstract and title are clear and reflect the gist of the manuscript.
The introduction and hypothesis are adequate as well.
Overall the article is well-written, however there are 2 queries.

The authors do not mention if the criteria used to identify the patients who are candidates for weaning was based on any society's guidelines, a set criteria, local guidelines, or selected at random by the authors.

Regarding the study design being retrospective, I have a major concern regarding the use of the custom biosignal collecting platform. From the knowledge of the ventilators and patient monitors used in ICU, the majority store the readings temporarily for less than 7days. 
So the obvious question here is that is the custom biosignaling collecting platform a part and parcel of the routine monitoring and data storage at the said hospital. If so, is this experimental technology approved for use in routine patient care?  If not approved, is this technology being used as a part of a larger trial being conducted for the validation of this technology? However, if the case is that this platform is not a part of routine critical care management and was used retrospectively, It is not clear how data was extracted from patients Jan 2019- Nov 2020, as by that time the memory of the devices would have reset.

Author Response

We truly appreciate all the constructive comments and invaluable suggestions from the editor and the reviewers.

We tried to respond point by point to the remarks addressed by the reviewer.

The full responses to the remarks is in the attached file.

Best regards.

Reviewer 2 Report

Thank you very much for giving me the opportunity to review the manuscript entitled “Biosignal-based digital biomarkers for prediction of ventilator weaning success”. The authors evaluated the differences in biosingal features between patients who successfully passed SBT and those who did not. Moreover, the authors demonstrated that the machine learning model using biosignal markers can predict weaning success more accurately than RSBI. The study approach and results are very interesting; however, the reviewer have several concerns as follows.

  1. In the univariate analysis comparing biosingal features between the weaning success and failure groups, the authors compared more than 30 valuables. Therefore, the p-value less than 0.05 cannot be regarded as statistically significant. The statistical significance levels must be adjusted for the multiple comparisons.
  2. The sample size is too small to evaluate the usefulness of the machine learning model. Moreover, the machine learning model was evaluated only in the single cohort. It is necessary to evaluate whether the model also works in other validation cohorts or not.
  3. Can multivariate logistic regression models including biosignal markers as covariates also distinguish weaning success and failure more accurately than the model including only RSBI?
  4. The journals and publication dates of many references are not presented.

Author Response

(The authors gave the same response as above.)

Reviewer 3 Report

This is a valuable paper. I have some comments.

1. What kind of pneumonia do the patients have? Were they not significantly different between the two groups?

2. If the target is narrowed down to patients with pneumonia, can RSBI+biosignal achieve the similar performance as this report?

3. In Table 1, please capitalize the first letter of pulmonary edema. Also, it is better to align the digits after the decimal point for the APCHE II score. Finally, references are not formatted according to the instructions for authors.

Author Response

(The authors gave the same response as above.)

Reviewer 4 Report

The study is interesting about an up-to-date topic of weaning from mechanical ventilation.

However, there is a lack of information within the methods. The parameters of all measured signals should be summarized (e.g. sample frequencies etc.). It is not clear why 62.5 Hz was used to downsample the signals. Some signals were probably waves and some were trend data with low sample rate frequency. 

The non-parametric test is mentioned within the paragraph statistical analysis however another part describes normally distributed data.

A description of the learning model is weak, there is no information about the teaching of the model (e. g. partition of data for the teaching and testing of the model).

Inequality of the groups (successful and control) is not discussed.

The representation of data in Tables 1 and 2 is misleading, e. g. blood gases in percentages - n(%). There is no discussion of negative values of evaluated parameters etc. 

 A number of parameters included in the model are not discussed.

Author Response

(The authors gave the same response as above.)

Reviewer 5 Report

The paper is related to predicting the outcome of weaning from mechanical ventilation. Although, the presentation of the paper is good but, in my opinion, the quantity of explanation in this paper is not enough to understand and follow the paper. For example, the machine learning model in explained in a couple of paragraphs with no mathematical modelling and explanation. same is the case with other parts of the paper such as statistical analysis. It is really important to include minimum level of mathematics and theoretical analysis to the paper to understand the concept. Also, add more explanation to the results as the tables are not explained enough.

Author Response

(The authors gave the same response as above.)

Round 2

Reviewer 2 Report

The manuscript has been significantly improved.

Author Response

We truly appreciate all the constructive comments and invaluable suggestions from the editor and the reviewers.

The full responses to the remarks is in the attached file.

Best regards.

Reviewer 5 Report

The paper has been revised and improved, however, my concern is still the same. The authors need to provide necessary contents about the ML algorithms adapted in this work. 

Author Response

(The authors gave the same response as above.)
